# Fluorescent Immunoassay with a Copper Polymer as the Signal Label for Catalytic Oxidation of *O*-Phenylenediamine

**DOI:** 10.3390/molecules27123675

**Published:** 2022-06-08

**Authors:** Yunxiao Feng, Gang Liu, Chunhuan Zhang, Jinrui Li, Yuanyuan Li, Lin Liu

**Affiliations:** 1College of Chemistry and Chemical Engineering, Pingdingshan University, Pingdingshan 467000, China; 2743@pdsu.edu.cn; 2College of Chemistry and Chemical Engineering, Henan University of Technology, Zhengzhou 450001, China; liugang08215@163.com; 3Henan Province of Key Laboratory of New Optoelectronic Functional Materials, Anyang Normal University, Anyang 455000, China; zch2142967813@163.com (C.Z.); lijinrui7022@163.com (J.L.)

**Keywords:** fluorescent immunoassay, peptide, copper, *O*-phenylenediamine, prostate specific antigen

## Abstract

This work suggested that Cu^2+^ ion coordinated by the peptide with a histidine (His or H) residue in the first position from the free N-terminal reveals oxidase-mimicking activity. A biotinylated polymer was prepared by modifying His residues on the side chain amino groups of lysine residues (denoted as K_H_) to chelate multiple Cu^2+^ ions. The resulting biotin-poly-(K_H_-Cu)_20_ polymer with multiple catalytic sites was employed as the signal label for immunoassay. Prostate specific antigen (PSA) was determined as the model target. The captured biotin-poly-(K_H_-Cu)_20_ polymer could catalyze the oxidation of *o*-phenylenediamine (OPD) to produce fluorescent 2,3-diaminophenazine (OPDox). The signal was proportional to PSA concentration from 0.01 to 2 ng/mL, and the detection limit was found to be eight pg/mL. The high sensitivity of the method enabled the assays of PSA in real serum samples. The work should be valuable for the design of novel biosensors for clinical diagnosis.

## 1. Introduction

Enzymes have been widely utilized as efficient signal labels in the construction of biosensors, including horseradish peroxidase (HRP), glucose oxidase (GOx) and alkaline phosphatase (ALP) [1,2]. However, the enzymatic signal amplification confronts low sensitivity because of the limited number of enzymes involved in the recognition event (generally one enzyme molecule in each event). The utilization of polymeric conjugates and nanomaterials for loading of numerous enzymes has proven to be an effective strategy to enhance the sensitivity [3,4]. However, some disadvantages, such as complicated preparation procedure, high cost, low stability and strict storage condition, seriously hamper the applications of enzyme labels [5]. Moreover, modification of enzymes on a solid surface may decrease the catalytic activity, and the large size of enzyme may cause the steric inhibition of antibody-antigen reaction. To resolve these shortcomings, the design and exploration of artificial enzyme mimics have aroused intensive interest in the field of bioassays. Recently, a series of nanomaterials defined as nanozymes such as peroxidase, catalase, superoxide dismutase, oxidase and so forth have been reported to possess unexpected and inherent enzyme-like catalytic abilities [5,6]. Despite the low price, high stability and easy preparation, nanozymes still suffer from some shortages such as low reproducibility, less specificity toward substrate, and plausible catalytic mechanism.

As a 3d transition metal, copper ion can accelerate a variety of reactions through a one- or two-electron pathway because of its wide range of accessible oxidation state (Cu^0^, Cu^I^, Cu^II^, and Cu^III^) [7,8]. Copper ion plays a critical role in biological systems such as metabolic reactions and enzyme catalysis. It has been reported that copper ion is accommodated in many metalloproteins (e.g., laccase, ceruloplasmin and ascorbate oxidase) and mainly acts as the active catalytic center to promote many redox reactions [9,10,11]. Mimicking the redox-active center of natural enzyme can greatly improve the catalytic ability and specificity of nanozymes based on the structure-activity relationship. Inspired by this phenomenon, many copper-functionalized metal, metal oxide and metal organic frameworks (MOFs)-based nanozymes have been synthesized through the coordination interaction and utilized to develop various biosensors. For example, Willner et al. suggested that Cu^2+^-modified carbon nitride nanoparticles, carbon dots, graphene oxide nanoparticles and MOFs possess HRP-mimicking activity and can be used as heterogeneous catalysts to promote the generation of chemiluminescence in the presence of luminole/H_2_O_2_ [12,13,14]. Meng et al. demonstrated that the Cu^2+^-containing carbon dots show laccase-like activity to catalyzing the oxidation of *p*-phenylenediamine [15]. Besides, some Cu^2+^-consisted nanomaterials show peroxidase-mimicking activity. For instance, MOF-818 with trinuclear copper centers was reported to exhibit catechol oxidase activity with excellent specificity [16]. Liu et al. found that MOFs formed by the coordination interaction between Cu^2+^ ions and guanosine monophosphates could be used as multicopper laccase mimicking nanozymes [10]. Zhao et al. suggested that copper hexacyanoferrate (Cu-HCF) nanozymes with single-site copper exhibit glutathione oxidase and peroxidase activities, which can initiate the cascade enzymatic reaction in tumor microenvironment [17]. These works suggest that copper ion can be used to design various catalysts or nanozymes with shape/size heterogeneity and high catalytic ability.

For a great majority of copper ion-containing proteins, histidine (His) residue located in different positions of peptide sequence has an important effect on the coordination of copper ion [18]. For example, peptides with a His residue in the first, second, or third position of N-terminal show different affinity capability and coordination mode towards copper ion [19,20,21,22,23,24]. We have reported that the peptide with a histidine (His) residue in the third position of N-terminal (denoted as ATCUN peptide) can limit the peroxidase and oxidase-like catalytic ability of copper ion, but the ATCUN peptide-copper complexes exhibit excellent activity for electrocatalytic water oxidation [25,26,27]. In this paper, we investigated the redox activity of copper ion coordinated by the peptide with a His residue in the first or second position from the N-terminus such as His-Ala (HA) and Ala-His (AH) (Figure 1). It was found that the HA-Cu^2+^ complexes could accelerate the electrochemical reduction of O_2_ and catalyze the chemical oxidation of ascorbic acid (AA) and o-phenylenediamine (OPD) by O_2_, but the AH peptide suppressed the electrochemical and catalytic activity of Cu^2+^. For this fact, His was modified on the side chain amino group of biotinylated polymeric lysine (denoted as K_H_) to obtain a biotin-poly-(K_H_)_20_ polymer. The polymer can be employed as a ligand to coordinate twenty copper ions for the formation of biotin-poly-(K_H_-Cu)_20_. The designed metal polymer was used as an artificial enzyme with multiple catalytic sites for the construction of biosensors. To demonstrate the analytical performances, prostate specific antigen (PSA), one of the most accurate and most extensively studied indicators of prostate cancer [28], was determined with the metal polymer as the signal label (Figure 2). Biotinylated secondary antibody (Ab_2_) was conjugated with the biotin-poly-(K_H_-Cu^2+^)_20_ polymer by using tetrameric streptavidin protein as the linker. The biotin-poly-(K_H_-Cu^2+^)_20_ polymer can catalyze the oxidation of OPD by O_2_ to produce fluorescent 2,3-diaminophenazine (OPDox) for signal readout.

## 2. Experimental

### 2.1. Chemicals and Reagents

Peptides were obtained from ChinaPeptides. Co., Ltd. (Shanghai, China). PSA and biotinylated secondary antibody (bio-Ab_2_) were provided by Linc-Bio Science Co. Ltd. (Shanghai, China). Immunoglobulin G (IgG), alpha-fetoprotein (AFP), human serum albumin (HSA) and enzyme-linked immunosorbent assay (ELISA) kits for PSA were purchased from Sigma-Aldrich. Thrombin was provided by Shanghai Yuanye Bio-Technology Co., Ltd. (Shanghai, China). Tris(hydroxymethyl)aminomethane (Tris) hydrochloride and OPD were obtained from Aladdin Corporation (Shanghai, China). Streptavidin and ascorbic acid (AA) were provided by Sangon biotech. Co., Ltd. (Shanghai, China). Human serum samples were obtained from the medical center of Anyang Normal University.

### 2.2. Voltammetric Characterization of Peptide-Cu^2+^

The redox behaviors of peptide-Cu^2+^ complexes in 50 mM pH 7.0 Tris buffer were characterized by cyclic voltammetry on a CHI 660E electrochemical workstation (CH instrument, Shanghai, China). Glass carbon electrode, platinum wire and Ag/AgCl electrode were used as the working, auxiliary and reference electrodes, respectively.

### 2.3. Absorption Measurement of AA

AA was diluted by Tris buffer containing a given concentration of Cu^2+^ or its complexes. The peptide concentration is slightly higher than that of Cu^2+^, ensuring that the free Cu^2+^ is negligible. The time-resolved UV-vis absorption was recorded with a fixed wavelength of 265 nm.

### 2.4. Absorption Measurement of AA

To evaluate the ability of free Cu^2+^ and peptide-Cu^2+^ complexes on the oxidation of OPD, a Cu^2+^-containing solution was mixed with the freshly prepared OPD solution at room temperature for 2 h. The mixture was then measured on a fluorescence spectrometer (Hitachi F-4600, Hitachi High-Tech, Japan).

### 2.5. PSA Detection

The Ab_2_-biotin-poly-(K_H_-Cu)_20_ conjugate was prepared by mixing streptavidin with equivalent bio-Ab_2_ at the concentration of 0.5 μM, followed by the addition of 1 μM biotin-poly-(K_H_)_20_. The fluorescent immunoassays of PSA were performed in a commercialized 96-well plate. A 100 μL PSA sample at a desired concentration was added to the plate well for 30-min incubation. After that, the well was washed with phosphate buffer, followed by the addition of 100 μL of Ab_2_-biotin-poly-(K_H_-Cu)_20_. After the formation of sandwich immunocomplex, the well was rinsed thoroughly with water and then 100 μL of Tris buffer containing 1 mM OPD was added to the well. After incubation for 1 h, the fluorescent signal was collected on the fluorescence spectrometer.

For the real sample assays, the serum samples were diluted ten times and the other procedures were the same as those for the standard sample detection. The concentrations were determined by three replicate measurements and deduced by the standard curve method.

## 3. Results and Discussion

### 3.1. Redox Behaviors of Cu^2+^ Complexes

The catalytic ability of copper complexes is greatly dependent upon the ligand nature and coordination mode. His residues in peptide play an important role in coordination of cooper ions. The peptide with a His residue in the first three N-terminal positions (His^1^, His^2^ and His^3^) shows unique Cu^2+^-binding property. Our group has investigated the redox property of Cu^2+^ coordinated by the ATCUN peptides with a His^3^ motif in the N-terminal position [25,26,27]. Such ATCUN-Cu^2+^ complexes exhibit the ability for electrocatalytic water oxidation with a potential at around 0.8 V. Herein, we investigated the redox activity of Cu^2+^ coordinated by the peptide with a His residue in the first or second position from the N-terminus. As shown in Figure 1A, the cyclic voltammogram (CV) of free Cu^2+^ exhibits a redox wave with a midpoint potential (E_1/2_) at around 0.05 V, which is attributed to the redox reaction of Cu^2+^/Cu^+^. The sharp anodic peak at −0.05 V is originated from the stripping of Cu^0^ [29]. The high background at the potential below −0.3 V can be attributed to the electrochemical reduction of O_2_ in solution [29]. For the HA-Cu^2+^ complex, an irreversible redox peak was observed in air-saturated solution. The redox peak became reversible when the solution was saturated by N_2_ (Figure 1B). The midpoint potential (−0.012 V) is more negative than that of free Cu^2+^. The results indicate that the electrogenerated HA-Cu^+^ can be rapidly oxidized into HA-Cu^2+^ by O_2_ in solution. No redox waves were observed for AH-Cu^2+^, indicating that the complex is redox inactive in the potential between 1.0 V and −0.4 V. A similar result was obtained for KH-Cu^2+^ (Figure 1C). However, when the His residue was modified on the side chain amino group of lysine, a couple of redox waves similar to that of HA-Cu^2+^ were observed. The redox waves become more irreversible at a lower scan rate (the inset in panel C), demonstrating that the generated K_H_-Cu^+^ can be immediately oxidized into K_H_-Cu^2+^ by oxygen. Similar CVs were observed for the Cu^2+^ complex formed with the biotin-poly-(K_H_)_20_ polymer (Figure 1D). The results suggest that Cu^2+^ coordinated by the peptide with a His residue in the first N-terminal position can facilitate the catalytic reduction of O_2_.

### 3.2. Catalytic Activity of Cu^2+^ Complexes

The redox potential of HA-Cu^2+^ and K_H_-Cu^2+^ complexes is higher than that of AA (−0.145 V) but lower than that of O_2_ [29]. From the thermodynamic viewpoint, the Cu^2+^ complex can be reduced by AA and the resulting Cu^+^ complex can be oxidized by O_2_, thus initiating the redox cycling between AA and O_2_. This process can be monitored by the absorbance change at 265 nm. As shown in Figure 2A, no significant change in the absorption intensity was observed for AA alone (curve 1), indicating that AA is stable at neutral pH. In the presence of a catalytic amount of free Cu^2+^ (curve 2), HA-Cu^2+^ (curve 3), K_H_-Cu^2+^ (curve 4), and biotin-poly-(K_H_-Cu^2+^)_20_ (curve 5), the absorption intensity decreased remarkably with time, indicating that these Cu^2+^ species can catalyze the oxidation of AA. The addition of AH-Cu^2+^ (curve 6) or KH-Cu^2+^ (curve 7) to the AA solution did not cause a significant decrease in the absorption intensity. Cu^2+^ can catalyze the oxidation of OPD to produce a yellow fluorescent product OPDox. Coordination of Cu^2+^ by some ligands such as quinolone, pyrophosphate, glyphosate, bleomycin, penicillamine and EDTA can inhibit the reaction between Cu^2+^ and OPD [30,31,32,33,34,35]. For this consideration, we investigated the reaction between the peptide-Cu^2+^ complexes and OPD. It was found that the peptides with a His residue in the second N-terminal position (AH and KH) depress the catalytic activity of Cu^2+^. Interestingly, the peptides with a His residue in the first N-terminal position such as HA, K_H_ and biotin-poly-(K_H_)_20_ promote the catalytic activity of Cu^2+^. These results indicate the Cu^2+^ complexes coordinated by the peptides with a His residue in the first N-terminal position exhibit oxidase-like activity for the oxidation of AA and OPD.

### 3.3. Sensitivity of Immunoassays

The detection of protein biomarkers is of great importance for early diagnosis and treatment of many diseases. Sandwich ELISA is the most commonly used method for protein detection in clinics. PSA is a well-known biomarker for prostate patients. To demonstrate the sensitivity of our strategy, PSA was determined by using biotin-poly-(K_H_-Cu^2+^)_20_ polymer as the label for signal amplification. As shown in Figure 2, the anti-PSA monoclonal antibody coated on the ELISA plate was used to capture PSA. The Ab_2_-biotin-poly-(K_H_-Cu^2+^)_20_ conjugate formed with biotinylated anti-PSA, SA and biotin-poly-(K_H_-Cu^2+^)_20_ polymer was employed for the signal readout. As shown in Figure 3A, the fluorescence intensity was gradually intensified with the increase of PSA concentration, indicating that a higher PSA concentration could lead to the capture of more PSA molecules and signal labels on the plate. Figure 3B depicts the relationship between the fluorescence intensity and PSA concentration. The relative standard deviations (RSDs) deduced from three replicate measurements for each concentration are all lower than 9%, which is indicative of acceptable repeatability and accuracy of this method. A linear fitting curve was obtained in the concentration of 0.01 and 2 ng/mL. The linear equation can be expressed as F = 245[PSA] (ng/mL) + 86. The detection limit was attained as low as 8 pg/mL, which is lower than that achieved by the ELISA kit with the enzyme-catalyzed signal amplification. Moreover, the sensitivity is comparable to those of the fluorescent immunoassays by the signal amplification of nanoenzymes or enzyme-loaded nanomaterials (Table 1). The high sensitivity can be attributed to the low background signal of the sandwich-like assay and the excellent catalytic ability of the Cu^2+^ polymer for OPD oxidation. The biotin-poly-(K_H_-Cu)_20_ polymer can be readily modified on the surface of nanomaterials through the avidin-biotin interaction. We believe that the sensitivity could be further improved by increasing the length of the polymer or by using nanomaterials to load large numbers of polymers.

### 3.4. Selectivity and Real Sample Assays

Selectivity is the basic requirement for the practical reliability of biosensors. The biosensor was challenged by determining low concentration of PSA and high concentration of interfering proteins. As depicted in Figure 4, no significant signals were observed for the tested interfering proteins, which is indicative of the excellent selectivity of the method. This can be attributed to the specific antigen-antibody interaction and the low non-specific adsorption of the polymer label. Female serum contains a negligible content of PSA. To demonstrate the anti-interference, female serum was spiked with PSA for assay. The result obtained in the female serum was consistent with that found in the buffer, indicating that the detection in the serum was interference-free. Encouraged by the result, PSA in three male serum samples was determined by this method and the commercialized ELISA kit (Table 2). The samples were diluted 10 times to ensure that the PSA concentration was in the linear range of the standard curve. The results obtained by our method are well coincident with those achieved by the commercial ELISA kit, demonstrating that the polymer label can be used to replace the enzyme label in clinical diagnosis. In contrast to the enzyme label, the polymer label shows the advantages of low manufacturing cost and ease of adjustability. Moreover, the RSDs deduced from three replicate measurements by this method are lower than those obtained by the ELISA, and no decrease in the signal was observed when the biotin-poly-(KH)20 polymer was kept at room temperature for at least five months, indicating the high thermodynamic stability of the polymer label.

## 4. Conclusions

In summary, we have investigated the electrochemical and catalytic activities of different peptide-Cu^2+^ complexes and designed a metal polymer for the development of immunoassays. The resulting biotin-poly-(K_H_)_20_ polymer was successfully used as an artificial enzyme label for PSA detection. Benefiting from the excellent analytical performances, the immunoassays achieved a low detection limit. The method was successfully used to detect PSA in real serum samples. The results are in agreement with those achieved by the commercial ELISA kit, indicating the great potential applications of the method for clinical assays of different biomarkers. Moreover, the polymer label exhibits the advantages of low cost and high stability and could be readily modified on the surface of nanomaterials to improve the sensitivity of immunoassays. We believe that this strategy can pave a new way to develop different sensing platforms for biomarker detection.

## Data Availability

Not applicable.

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
