# Peer review of "Fluorescent Immunoassay with a Copper Polymer as the Signal Label for Catalytic Oxidation of *O*-Phenylenediamine"

_molecules, 2022, doi:10.3390/molecules27123675_

Round 1

Reviewer 1 Report

The manuscript entitled "Fluorescent immunoassay with a copper polymer as the signal label for catalytic oxidation of o-phenylenediamine" written as per journal style. However, since this is new analytical procedure i suggest inclusion of more analytical data to support your results. for example. Precision and accuracy; stability, any matrix effect. 

A new method utilising Fluorescent immunoassay with a copper polymer as the signal label for catalytic oxidation of
o-phenylenediamine, has been developed and reported. The article is written as presented in the journal style with recent references. A compared to the other methods, the present method is comparable. The conclusion is drawn on the basis of the present data-set.  Its application in the estimation of PSA is  reported. However, a few minor corrections are required.  1. The data in Table 2 requires statistical analysis and p values. 2. mention how many replicate were analysed 3, 5 ( as the data is presented as mean+sd) 3. Is there any matrix effect? 4. Justify the analytical data and present precision and accuracy data for the clinical analysis 5. Please justify the results with standard addition and recovery methods.

Author Response

We thank the reviewer for his/her positive and constructive comments: “The manuscript entitled "Fluorescent immunoassay with a copper polymer as the signal label for catalytic oxidation of o-phenylenediamine" written as per journal style. However, since this is new analytical procedure i suggest inclusion of more analytical data to support your results. for example. Precision and accuracy; stability, any matrix effect. A new method utilising Fluorescent immunoassay with a copper polymer as the signal label for catalytic oxidation of o-phenylenediamine, has been developed and reported. The article is written as presented in the journal style with recent references. A compared to the other methods, the present method is comparable. The conclusion is drawn on the basis of the present data-set. Its application in the estimation of PSA is reported. However, a few minor corrections are required.

Comment 1: The data in Table 2 requires statistical analysis and p values.

Response: The results were deduced from three replicate measurements and the relative standard deviations (RSDs) for this method and the ELISA were discussed in the revised manuscript.

Comment 2:mention how many replicate were analysed 3, 5 (as the data is presented as mean+sd)

Response: The results were deduced from at least three replicate measurements and the absolute errors are shown as the error bars. We have discussed the relative standard deviations (RSDs) for the assays of the standard and serum samples.

Comment 3:Is there any matrix effect?

Response: Female serum contains a negligible content of PSA. To demonstrate the anti-interference, female serum was spiked with PSA for assay. The result obtained in the female serum was consistent with that found in the buffer, indicating that the detection in serum was interference-free. We have added the result and discussion in the revised manuscript.

Comment 4:Justify the analytical data and present precision and accuracy data for the clinical analysis.

Response: The precision and accuracy can be reflected by the relative standard deviations (RSDs). The RSDs deduced from three replicate measurements by this method are lower than those obtained by the ELISA, and no decrease in the signal was observed when the biotin-poly-(KH)20 polymer was kept at room temperature for at least five months, indicating the high thermodynamic stability of the polymer label. We have added the discussion in the manuscript.

Comment 5: Please justify the results with standard addition and recovery methods.

Response: The results for real sample assays were obtained by the standard curve method. We have provided the information in the revised manuscript.

Reviewer 2 Report

The article made a good impression on me. The introduction provides the necessary background and demonstrates the importance of this research. The article is well-written from a scientific point of view and the data presented are properly discussed. The results obtained are important and will be of interest to readers of the journal Molecules. The material in the article, including the figures, is well designed, and, in my opinion, the article can be accepted for publication in Molecules after minor revision. Minor shortcomings relate to the statistical analysis.

Author Response

We thank the reviewer for his/her positive comments: The article made a good impression on me. The introduction provides the necessary background and demonstrates the importance of this research. The article is well-written from a scientific point of view and the data presented are properly discussed. The results obtained are important and will be of interest to readers of the journal Molecules. The material in the article, including the figures, is well designed, and, in my opinion, the article can be accepted for publication in Molecules after minor revision. Minor shortcomings relate to the statistical analysis.” We have checked the manuscript carefully and discussed the relative standard deviations (RSDs) for the assays of different concentrations of samples.